# Pelagic cnidarian assemblages show range-edge effect at the boundary of ocean surface, as illustrated in the case of the Amazon River Plume

**Everton Giachini Tosetto** [1,2,3]*, **Sigrid Neumann-Leitão**[3], **Moacyr Araujo**[3,4], **Arnaud Bertrand** [1,2,3,5], **Miodeli Nogueira Júnior**[6]

**1** Institut de Recherche pour le Développement, Sète, France, **2** MARBEC, CNRS, IFREMER, IRD, Univ Montpellier, Sète, France, **3** Departamento de Oceanografia, Universidade Federal de Pernambuco, Recife, Brazil, **4** Brazilian Research Network on Global Climate Change (Rede CLIMA), São José dos Campos, Brazil, **5** Departamento de Pesca e Aquicultura, Universidade Federal Rural de Pernambuco, Recife, Brazil, **6** Departamento de Sistemática e Ecologia, Universidade Federal da Paraíba, João Pessoa, Brazil

* evertontosetto@hotmail.com

**Data Availability Statement:** All relevant data are within the manuscript and its Supporting Information files.

## Abstract

The neuston layer represents a complex community inhabiting the interface where oceanographic and atmospheric processes interact. Here, our aim was to compare patterns in the distribution and abundance of cnidarian assemblages observed in the neuston to parallel patterns previously observed in epipelagic waters along the spread of the Amazon River Plume over the Western Equatorial Atlantic, to test if the neuston reflects the patterns of the overall community whose core of distribution is located in epipelagic waters or are shaped by specific surface processes. The results show that both initial hypothesis were false. Instead, the cnidarian assemblages showed range-edge effect at the major ecotone placed at the interface between ocean and atmosphere. I.e., when proximate to the superior limits of their three-dimensional geographic ranges, represented here by the neuston, the population of most observed species occur in lower abundance. Specifically at the portion of the continental shelf with influence of the Amazon River Plume, the range-edge effect seems to be more prominent. Such results suggests the core of the cnidarian populations inhabiting this habit may lie in the deeper hypoxic waters beneath the plume. In conclusion, due the marked vertical structure observed here, proper evaluations of spatial patterns in the structure of pelagic cnidarian communities should preferentially be grounded on stratified sampling.

## Introduction

The neuston comprises the biological community inhabiting the surface layer of aquatic systems at the interface with the atmosphere [1, 2]. In the ocean, it is a critical zone for marine biota, where in addition to physical oceanographic processes and surface water conditions, the

**Funding:** The author(s) received no specific funding for this work.

**Competing interests:** The authors have declared that no competing interests exist.

system is influenced by atmospheric variability. Thus, parameters such as light intensity, wind stress, turbulence and air temperature contribute in shaping the distribution of organisms [1]. The surface layer is also an accumulation zone of terrestrial debris, pollutants, micro and macro plastics and river freshwater run-off [3–5]. All these features increase the complexity of the neustonic habitat, and consequently the dynamics of its assemblages.

Despite its thinness, the neustonic layer covers 71% of the surface of the planet, representing one of the most ubiquitous habitats [6]. It plays a major role in global biogeochemical processes, including $CO_2$ assimilation and production of marine snow, which contribute to the sinking of carbon to deeper layers [7, 8]. It is also a key environment for the life cycle and reproduction of numerous marine organisms, with high abundance and diversity of larval and juvenile stages [9–11]. In addition, the neustonic assemblage is an important food source for fish and other pelagic groups as well as seabirds [12–14].

In comparison to traditional zooplankton sampling, sampling the neuston have several advantages. First, it does not require the use of vertical or oblique deep hauls, which allows for sampling with any type of platform, including sailing boats. This is advantageous as it is faster, less expensive and allows for the collection of continuous data. Therefore, if the neuston reflects the patterns of the overall community whose core of distribution is located in epipelagic waters, then using the neustonic portion of the pelagic organisms would be a promising approach in ecological studies. Alternatively, in environments vertically structured, if the distribution of the neustonic community is directly shaped by specific surface processes that do not affect deeper communities, then the neustonic organisms would be an indicator of such processes.

To our knowledge, the assumption that the neustonic communities are representative of the deeper epipelagic community patterns, as well as the specific surface patterns affect their distribution was never addressed. Therefore, to elucidate the question, we propose to use pelagic cnidarians as a study model. While fish and other organisms sporadically visit the neuston to feed [12, 13], organisms with less mobility such as small cnidarians may permanently inhabit it, with a few floating species being characteristic of the environment [15]. However, studies subjecting zooneuston tend to focus on upper taxa [16–18] or dominant groups such as copepods [19]. Considering pelagic cnidarians, most research has focused on the water column, a few to hundreds of meters below the neustonic layer. These pelagic cnidarian communities have faced increased scientific attention due a set of factors. First, they play a significant role both as predators and preys, contributing to the functioning of marine food webs [20–22]. Cnidarians also have strong responses to physical, biogeochemical and ecological processes of ocean (e.g. distribution of water masses, passive dispersal by currents and production regimes) [23–25], what makes them good model organisms to comprehend the relation between these processes and marine life. Finally, cnidarians have received public and scientific attention due the controversial perception that jellyfish population blooms are increasing globally along the Anthropocene [26–29].

Here, we use data collected in the neustonic layer and compare it to parallel patterns observed in epipelagic (0–200 m depth) communities [30] along the spread of the Amazon River Plume (hereafter ARP), over the Western Equatorial Atlantic Ocean, a region exhibiting high horizontal and vertical structuring. The massive discharge of the Amazon River creates a surface plume (ARP) of low-salinity, high nutrients, and suspended and dissolved materials which spreads through the surface layer reaching the Caribbean Sea and North Atlantic [5]. Such characteristics shapes marine habitats with a thin highly productive surface layer compressed by a deeper oxygen minimum zone both over the shelf and in the open ocean. This layer apparently is particularly advantageous to cnidarians [30], which have low metabolic rates, being able to survive in hypoxic zones [31, 32]. Therefore, the ARP was observed as the

main driver shaping the distribution of epipelagic planktonic cnidarians in both neritic and oceanic habitats from the Western Equatorial Atlantic, resulting in high species richness and abundance [30].

Thus, in this study we evaluated the spatial structure of the neustonic cnidarian community from the continental shelf and adjacent oceanic waters of Northern Brazil in the Equatorial Atlantic and their responses to spatial changes in the physical environment, comparing the patterns concomitantly observed in the epipelagic communities. This opens the opportunity for us to test for the two alternative hypotheses: (1) the neustonic cnidarian communities are affected by the ARP and its oxygen minimum zone in a similar way than epipelagic communities, being therefore, representative of one whole epipelagic community, or (2) the effects observed in epipelagic communities are more pronounced to organisms in the neuston, since it is the layer under higher influence of the ARP, resulting in communities with distinct composition and abundance in both layers.

## Materials and methods

### Study area

The study area was along the North Brazilian continental shelf between the Amazon and Oyapok river mouths and in Equatorial Atlantic oceanic waters, ranging between 8˚N, 51˚W and 3.5˚S, 37˚W (Fig 1). In the area, the continental shelf reaches up to 300 km wide and the shelf

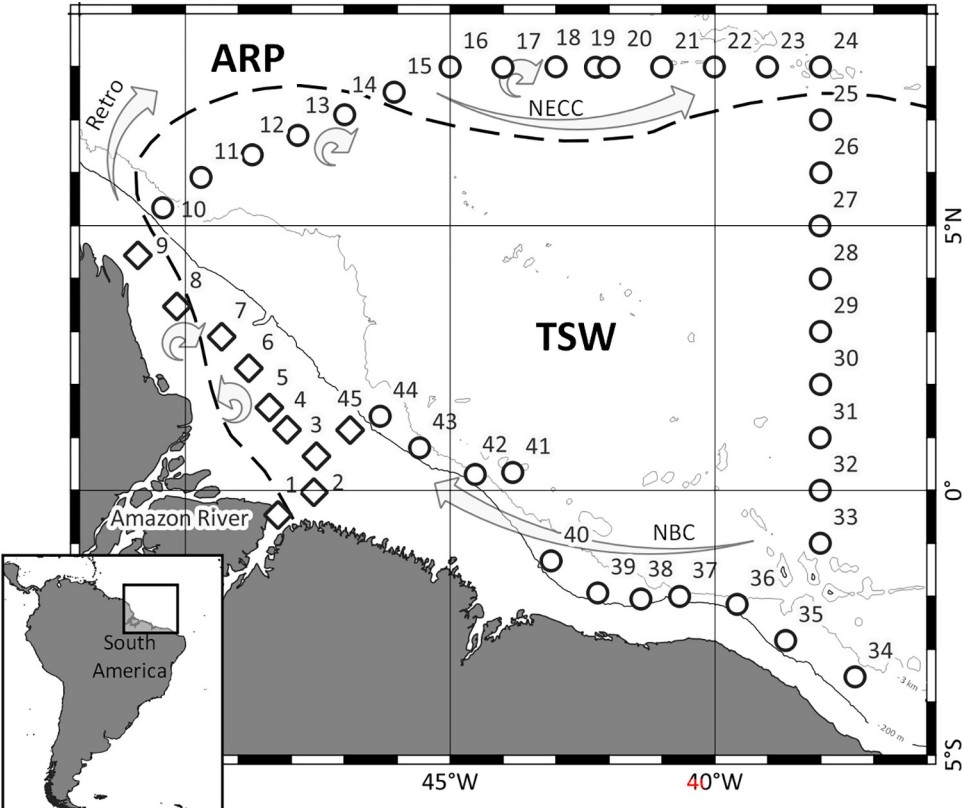

**Fig 1. Geographic location of the study area in the North Brazilian continental shelf and adjacent Western Equatorial Atlantic Ocean, showing the sampled stations (diamonds = shelf, circles = open ocean) and mesocale processes occurring in October 2012.** ARP = Amazon River plume; TSW = Tropical Surface Water; NBC = North Brazilian Current; RETRO = North Brazilian Current retroflection; NECC = North Equatorial Counter Current. Dashed line = 35 isohaline indicating surface limits of the ARP.

break occurs around 120 m depth [33, 34]. The reach of the ARP in the area is mainly affected by its discharge, the North Brazilian Current (NBC) and wind fields [5, 34–38]. Three general patterns occur throughout the year. Between January and April, the ARP flow continuously northwest along the Brazilian coast carried mainly by NE winds. From April to July the ARP reaches the Caribbean region due to the higher discharge, NBC transport and SE winds. From August to December, the retroflection of the NBC, around 5˚N and 10˚N, disperses the ARP to the east feeding the North Equatorial Counter Current (NECC) [5, 39].

## Sampling

Data were collected during the oceanographic cruise Camadas Finas III, aboard the research vessel NHo. Cruzeiro do Sul—H38 (DHN/Brazilian Navy) in the western equatorial Atlantic (Fig 1). The cruise was performed in October 9–31, 2012, a period when the retroflection of the NBC disperses the plume to the east feeding the NECC [5]. Zooneuston samples were obtained at 45 stations along the track of the cruise with a David-Hempel aluminum catamaran (Hydro-Bios, https://www.hydrobios.de/images/datasheets/438%20210_Neuston.pdf) equipped with two superposed nets with rectangular mouth (30x15 cm each one) and a 500 μm mesh. The upper net was adjusted exactly centered at the air-water interface, sampling the epineuston layer from the surface to 7.5 cm depth, while the lower net sampled the hyponeuston layer from 7.5 cm to 22.5 cm depth. The catamaran was hauled during 20 minutes at a speed of 2–3 knots each station. The lower net was fitted with a flowmeter (Hydro-Bios) to estimate the volume filtered during each trawl. The zooneuston samples were fixed with 4% formaldehyde buffered with sodium tetraborate (0.5 g.l$^{-1}$).

In laboratory, whole samples were analysed under stereomicroscope and cnidarian specimens were identified [mainly following 40–42] and counted. A general taxonomic overview of neustonic cnidarians of the study area was previously reported [43]. Abundances were standardized as number of individuals per 100 m$^{-3}$ for medusae and number of colonies per 100 m$^{-3}$ for siphonophores. For calycophorans, the number of anterior nectophores was used for estimating the polygastric stage abundance, and eudoxid bracts for the eudoxid stage abundance [e.g. 44, 45]. For physonects and the calycophora *Hippopodius hippopus*, number of colonies were roughly estimated by dividing the number of nectophores by 10 [46].

Salinity, temperature (˚C), density (σ$_t$), dissolved oxygen (mgL$^{-1}$) and fluorescence vertical profiles, characterization of water masses, current speed and directions and presence of cyclonic and antyciclonic eddies were obtained for the sampling period. Methods and results were described in Tosetto et al. [30]. Main mesoscale processes observed in the area are shown in Fig 1.

## Data analysis

To dampen effects of dominant species, in all analyses abundance data was transformed by log (x + 1). Station 1 was excluded from all analysis since it was the only station sampled inside the estuary. A Permutational Multivariate Analysis of Variance [PERMANOVA; 47] was used to test for differences in the cnidarian community structure between epineuston and hiponeuston. Since no significant difference were observed (PseudoF = 1.594, P = 0.107) we considered both samples from each station as replicates.

Spatial patterns in neustonic cnidarian community abundance were identified by hierarchical cluster analysis using Bray–Curtis similarity matrix. The validity of the groups defined by the cluster analysis was tested though SIMPROF test (5% significance level). A Similarity Percentage (SIMPER) analysis was performed to identify representative species and their contribution to similarity within the groups defined by the cluster analysis [48].

To identify associations between representative neustonic cnidarian taxa (species occurring in more than 21 stations and species with high abundance in few stations) and the environment, a constrained ordination analysis was performed. Detrended Canonical Correspondence Analysis (DCCA) revealed a small length of variable gradients (<3), indicating that a linear method was appropriate to use on this occasion, and thus Redundancy Analysis (RDA) was selected [49]. Predominant current was included as dummy categorical explanatory variable. Bottom depth, max value of fluorescence (as an indirect measure of biological productivity) in the first 200 m of the water column, and surface temperature and salinity were included as continuous explanatory variables.

Distribution maps were produced in QGIS 3.10 [50]. Cluster, SIMPROF, SIMPER and PERMANOVA analysis were performed in Primer v.6+PERMANOVA [48]. DCCA and RDA were performed in CANOCO 4.5 [49].

## Results

### Species composition

A total of 55 cnidarian taxa were observed in the neustonic layer of the study area, corresponding to one scyphomedusae, 23 hydromedusae (hereafter pelagic hydroids of Anthoathecate are included in this category for convenience) and 31 siphonophores (Table 1). Furthermore, unidentifiable cerinula, ephyrae and athorybia larval forms were found. *Liriope tetraphylla* was the most frequent hydromedusae, present in 55.6% of the samples, followed by *Porpita porpita* (26.7%), *Aglaura hemistoma* (24.4%) and *Cytaeis* sp.3 (22.2%, Table 1). *L. tetraphylla* also dominated in abundance, representing 56.1% of all hydromedusae. Other representative hydromedusae were *Cytaeis* sp.3 (17.8%) and *Cunina octonaria* (11.5%). Among Siphonophores, the most frequent were *Chelophyes appendiculata* (67.8%), *Diphyes bojani* (61.1%), *Bassia bassensis* (57.8%), *Abylopsis eschscholtzii* (56.7%) and *Diphyes dispar* (50%; Table 1). When considering abundance, *C. appendiculata* also dominated (32%), followed by *D. bojani* (19.6%), *D. dispar* (12.1%), *B. bassensis* (9%) and *A. eschscholtzii* (7.2%).

### Spatial distribution patterns

Hydromedusae diversity was generally low both over the continental shelf (1.8 ± 2.3) and in the open ocean (1.8 ± 1.5). However, some samples under the influence of the ARP in both environments presented slightly higher diversity, reaching 9 species in station 8 (Fig 2A). Otherwise, hydromedusae total abundance over the continental shelf peaked outside the ARP. In the open ocean, stations under influence of the ARP and in the retroflection area (particularly station 10) presented higher abundances, which were generally lower in the remaining open ocean (Fig 2B). Number of siphonophore species was generally high all over the oceanic province (Fig 2C), averaging 6.8 ± 3 species per sample. Over the continental shelf, stations under influence of the ARP and station 45, near the shelf break, presented slightly higher siphonophore species (Fig 2C). Siphonophores total abundance was high throughout the oceanic province as well (Fig 2D). Over the continental shelf no clear pattern was observed, with stations inside and outside the ARP presenting high abundances (Fig 2D).

*Liriope tetraphylla*, *Diphyes dispar* and *Diphyes bojani* were the three dominant species present both over the continental shelf and in the open ocean (Fig 3A–3C). *L. tetraphylla* was the main species responsible for hydromedusae abundance pattern over the continental shelf. There, the species was quite more abundant outside the ARP, particularly at stations were cyclonic and anti-cyclonic eddies were present, reaching 810.1 ind. 100 m$^{-3}$ at station 6 (Fig 3A). In the open ocean *L. tetraphylla* was widespread over the area, however higher abundances were observed in the retroflection region, were the species peaked in station 10 (855.9

**Table 1. Basic statistics of neustonic cnidarian species from neritic and oceanic provinces in the Western Equatorial Atlantic Ocean.** Mean abundance (ind. 100 m$^{-3}$) and standard deviation, range of abundance, frequency of occurrence (f; considering both provinces, n = 90 samples) and temperature and salinity ranges.

| Species | Neritic | | Oceanic | | f | Temperature | Salinity |
|---|---|---|---|---|---|---|---|
| | Mean ± SD | Range of non-zero abundances | Mean ± SD | Range of non-zero abundances | | | |
| **"Hydromedusae"** | | | | | | | |
| *Liriope tetraphylla* (Chamisso & Eysenhardt, 1821) | 106.52 ± 228.22 | 3.09–810.1 | 29.1 ± 129.86 | 1.62–855.85 | 55.56 | 26.26–29.69 | 18.18–36.64 |
| *Porpita porpita* (Linnaeus, 1758) floating colonies | 0.39 ± 1.72 | 7.71–7.71 | 6.42 ± 19.18 | 1.67–130.14 | 26.67 | 26.26–29.78 | 32.78–36.35 |
| *Aglaura hemistoma* Péron & Lesueur, 1810 | 5.21 ± 12.91 | 2.39–46.26 | 1.31 ± 3.31 | 1.36–16.69 | 24.44 | 26.38–29.69 | 31.58–37.06 |
| *Cytaeis* sp.3 | - | - | 18.89 ± 65.36 | 1.76–344.86 | 22.22 | 26.65–29.69 | 32.78–36.3 |
| *Cunina octonaria* McCrady, 1859 | 0.17 ± 0.69 | 0.38–3.09 | 12.21 ± 92.87 | 2.82–776.07 | 8.89 | 27.81–29.6 | 31.58–36.48 |
| *Annatiara affinis* (Hartlaub, 1914) | - | - | 0.47 ± 1.89 | 2.29–13.24 | 6.67 | 27.63–29.55 | 32.55–36.29 |
| *Rhopalonema velatum* Gegenbaur, 1857 | 0.19 ± 0.86 | 3.86–3.86 | 0.14 ± 0.89 | 3.34–6.68 | 3.33 | 26.61–27.99 | 36.08–36.26 |
| *Clytia* spp. | 0.35 ± 1.08 | 3.09–3.86 | 0.02 ± 0.16 | 1.36–1.36 | 3.33 | 26.61–29.04 | 31.58–36.26 |
| *Aequorea macrodactyla* (Brandt, 1835) | 0.96 ± 3.51 | 3.86–15.42 | 0.01 ± 0.12 | 0.99–0.99 | 3.33 | 26.52–26.61 | 36.26–36.27 |
| *Corymorpha gracilis* (Brooks, 1883) | - | - | 0.09 ± 0.56 | 1.81–4.32 | 2.22 | 27.89–28.4 | 35.94–36.12 |
| *Laodicea undulata* (Forbes & Goodsir, 1853) | - | - | 0.05 ± 0.32 | 1.19–2.39 | 2.22 | 26.5–26.5 | 36.31–36.31 |
| *Pegantha triloba* Haeckel, 1879 | - | - | 0.52 ± 3.86 | 4.63–32 | 2.22 | 27.63–28.07 | 36–36.29 |
| *Malagazzia carolinae* (Mayer, 1900) | 0.31 ± 1.38 | 6.18–6.18 | 0.02 ± 0.14 | 1.21–1.21 | 2.22 | 27.81–29.55 | 31.58–32.55 |
| *Persa incolorata* McCrady, 1859 | 1.39 ± 4.96 | 6.18–21.62 | - | - | 2.22 | 27.81–27.81 | 31.58–31.58 |
| *Eirene viridula* (Péron & Lesueur, 1810) | 4.1 ± 15.12 | 15.44–66.63 | - | - | 2.22 | 27.81–28.23 | 18.18–31.58 |
| Eirenidae sp. | - | - | 0.04 ± 0.3 | 2.52–2.52 | 1.11 | 26.38–26.38 | 36.3–36.3 |
| *Mitrocomella* sp. | - | - | 0.1 ± 0.82 | 6.86–6.86 | 1.11 | 27.63–27.63 | 36.29–36.29 |
| *Pegantha laevis* H.B. Bigelow, 1909 | - | - | 0.03 ± 0.29 | 2.4–2.4 | 1.11 | 27.91–27.91 | 36.17–36.17 |
| *Stauridiosarsia producta* (Wright, 1858) | - | - | 0.04 ± 0.36 | 3.03–3.03 | 1.11 | 26.95–26.95 | 36.3–36.3 |
| *Velella velella* (Linnaeus, 1758) floating colonies | - | - | 0.02 ± 0.21 | 1.75–1.75 | 1.11 | 26.65–26.65 | 36.3–36.3 |
| Bougainvilliidae sp. | 0.15 ± 0.69 | 3.09–3.09 | - | - | 1.11 | 27.81–27.81 | 31.58–31.58 |
| *Octophialucium haeckeli* (Vannucci & Soares Moreira, 1966) | 0.15 ± 0.69 | 3.09–3.09 | - | - | 1.11 | 27.81–27.81 | 31.58–31.58 |
| *Eucheilota* sp. | 0.31 ± 1.38 | 6.18–6.18 | - | - | 1.11 | 27.81–27.81 | 31.58–31.58 |
| **Scyphomedusae** | | | | | | | |
| *Nausithoe maculata* Jarms, 1990 | - | - | 0.05 ± 0.4 | 3.35–3.35 | 1.11 | 29.69–29.69 | 34.4–34.4 |
| Ephyrae | 0.25 ± 1.11 | 4.95–4.95 | 1.79 ± 11.74 | 1.36–97.01 | 7.78 | 28.05–29.69 | 34–36.51 |
| **Siphonophorae** | | | | | | | |
| *Chelophyes appendiculata* (Eschscholtz, 1829) | 4.55 ± 13.08 | 6.18–46.26 | 38.41 ± 71.15 | 1.07–391.37 | 67.78 | 26.21–29.78 | 31.58–36.35 |
| *Diphyes bojani* (Eschscholtz, 1825) | 5.42 ± 15.78 | 1.56–61.69 | 22.74 ± 40.83 | 0.64–179.61 | 61.11 | 26.21–29.69 | 31.05–36.51 |
| *Bassia bassensis* (Quoy & Gaimard, 1833) | 0.31 ± 1.38 | 6.18–6.18 | 11.04 ± 19.15 | 0.64–120.39 | 57.78 | 26.21–29.69 | 31.58–36.35 |
| *Abylopsis eschscholtzii* (Huxley, 1859) | 0.4 ± 1.72 | 0.38–7.71 | 8.76 ± 11.7 | 1.67–55.27 | 56.67 | 26.21–29.69 | 32.55–36.48 |
| *Diphyes dispar* Chamisso & Eysenhardt, 1821 | 12.8 ± 24.31 | 6.24–81.75 | 11.37 ± 31.7 | 0.87–203.34 | 50 | 26.61–29.78 | 31.05–36.51 |
| *Sulculeolaria chuni* (Lens & van Riemsdijk, 1908) | 2.12 ± 6.56 | 19.28–23.13 | 4.2 ± 12.09 | 0.64–79.44 | 42.22 | 26.38–29.78 | 32.55–36.31 |
| *Abylopsis tetragona* (Otto, 1823) | 0.39 ± 1.72 | 7.71–7.71 | 3.48 ± 5.8 | 1.28–30.27 | 40 | 26.21–29.69 | 34–36.35 |
| *Agalma okenii* Eschscholtz, 1825 | - | - | 2.24 ± 4.73 | 0.64–30.98 | 37.78 | 26.26–29.69 | 32.55–36.35 |
| *Nanomia bijuga* (Delle Chiaje, 1844) | 1.21 ± 2.32 | 2.48–6.24 | 1.18 ± 2.38 | 1.41–11.03 | 28.89 | 27.81–29.78 | 31.05–36.51 |
| *Eudoxoides spiralis* (Bigelow, 1911) | - | - | 2.14 ± 4.86 | 0.66–22.62 | 22.22 | 26.21–28.26 | 36–36.35 |

*(Continued)*

**Table 1.** (Continued)

| Species | Neritic | | Oceanic | | f | Temperature | Salinity |
|---|---|---|---|---|---|---|---|
| | Mean ± SD | Range of non-zero abundances | Mean ± SD | Range of non-zero abundances | | | |
| *Muggiaea kochii* (Will, 1844) | - | - | 0.87 ± 1.91 | 0.66–8.73 | 22.22 | 26.21–28.45 | 31.05–36.35 |
| *Lensia campanella* (Moser, 1917) | 0.19 ± 0.86 | 3.86–3.86 | 0.37 ± 1.08 | 1.21–4.63 | 11.11 | 26.49–29.55 | 32.55–36.3 |
| *Eudoxoides mitra* (Huxley, 1859) | 0.39 ± 1.72 | 7.71–7.71 | 0.33 ± 1.01 | 1.2–6.03 | 11.11 | 26.61–29.6 | 32.55–36.26 |
| *Ceratocymba leuckartii* (Huxley, 1859 | - | - | 0.46 ± 1.49 | 1.07–8.17 | 10 | 26.95–29.6 | 32.78–36.3 |
| *Sulculeolaria turgida* (Gegenbaur, 1854) | - | - | 0.37 ± 1.09 | 0.64–5.18 | 10 | 26.92–29.55 | 32.55–36.29 |
| *Cordagalma ordinatum* (Haeckel, 1888) | 0.19 ± 0.86 | 3.86–3.86 | 0.43 ± 1.49 | 1.67–9.05 | 10 | 26.61–29.52 | 34.23–36.3 |
| *Agalma elegans* (Sars, 1846) | - | - | 0.4 ± 1.39 | 0.99–9.14 | 8.89 | 26.52–27.99 | 36.08–36.3 |
| *Sulculeolaria monoica* (Chun, 1888) | - | - | 1.92 ± 11.34 | 1.36–88.5 | 6.67 | 28.26–29.55 | 32.55–36.29 |
| *Solmundella bitentaculata* (Quoy & Gaimard, 1833) | 0.19 ± 0.86 | 3.86–3.86 | 0.66 ± 2.82 | 3.23–18.54 | 6.67 | 26.61–29.48 | 33.51–36.29 |
| *Physophora hydrostatica* Forsskål, 1775 | - | - | 0.19 ± 0.84 | 1.2–4.57 | 4.44 | 27.63–29.38 | 34–36.29 |
| *Enneagonum hyalinum* Quoy & Gaimard, 1827 | 5.27 ± 12.46 | 15.6–49.41 | | | 4.44 | 27.81–27.88 | 31.05–31.58 |
| *Physalia physalis* (Linnaeus, 1758) | - | - | 0.13 ± 0.64 | 1.98–3.68 | 3.33 | 26.52–29.78 | 34.23–36.27 |
| *Abyla* sp. | - | - | 0.05 ± 0.33 | 1.36–2.4 | 2.22 | 27.91–29.04 | 35.45–36.17 |
| *Athorybia rosacea* (Forsskål, 1775) | - | - | 0.12 ± 0.7 | 3.68–4.63 | 2.22 | 28.07–29.52 | 34.23–36 |
| *Halistemma rubrum* (Vogt, 1852) | - | - | 0.1 ± 0.61 | 2.29–4.63 | 2.22 | 27.63–28.07 | 36–36.29 |
| *Lensia conoidea* (Keferstein & Ehlers, 1860) | - | - | 0.12 ± 0.8 | 1.73–6.47 | 2.22 | 27.89–28.45 | 35.58–36.12 |
| *Lensia cossack* Totton, 1941 | - | - | 0.05 ± 0.31 | 1.21–2.29 | 2.22 | 27.63–29.55 | 32.55–36.29 |
| *Sulculeolaria biloba* (Sars, 1846) | - | - | 0.05 ± 0.3 | 0.99–2.32 | 2.22 | 26.52–28.07 | 36–36.27 |
| *Lensia subtilis* (Chun, 1886) | 0.25 ± 1.11 | 4.95–4.95 | 0.04 ± 0.36 | 3.03–3.03 | 2.22 | 26.95–28.05 | 36.3–36.51 |
| *Hippopodius hippopus* (Forsskål, 1776) | - | - | 0.02 ± 0.14 | 1.21–1.21 | 1.11 | 29.55–29.55 | 32.55–32.55 |
| *Lensia subtiloides* (Lens & van Riemsdijk, 1908) | - | - | 0.01 ± 0.08 | 0.64–0.64 | 1.11 | 26.92–26.92 | 36.28–36.28 |
| Athorybia larvae | - | - | 0.05 ± 0.4 | 3.35–3.35 | 1.11 | 29.69–29.69 | 34.4–34.4 |
| **Anthozoa** | | | | | | | |
| Cerinula | - | - | 0.54 ± 2.7 | 0.66–20.18 | 5.56 | 26.21–26.5 | 36.3–36.33 |

ind. 100 m$^{-3}$; Fig 3A). *D. dispar* was more abundant in stations under influence of the ARP or in its vicinity, both over the continental shelf and in the open ocean (Fig 3B). *D. bojani* was more abundant in stations under influence of the ARP or in its vicinity in the open ocean and scattered distributed over the continental shelf (Fig 3C). *Enneagonun hyalinun*, *Muggiaea kocchii* and *Persa incolarata* were the more abundant species present exclusively in the area under influence of the ARP over the continental shelf (Fig 3D–3F). *C. appendiculata*, *Bassia bassensis A. eschscholtzii* and *Porpita porpita* were widespread in the open ocean both in the area under influence of the ARP and outside it (Fig 3G, 3J–3I). *Cytaeis* sp.3, was absent over the continental shelf. In the open ocean the species was scattered distributed through the area, however its abundance was quite higher in stations under influence of the ARP (Fig 3H). *C. octonaria* occurred in high abundance exclusively at station 10 where it reached 776.1 ind. 100 m$^{-3}$, only occasional catches of the species occurred in other stations (Fig 3I).

## Community structure

The cluster analysis depicted two main groups (A and B) with less than 30% similarity between each other (Fig 4A). Group A was mostly composed by stations over the continental shelf and

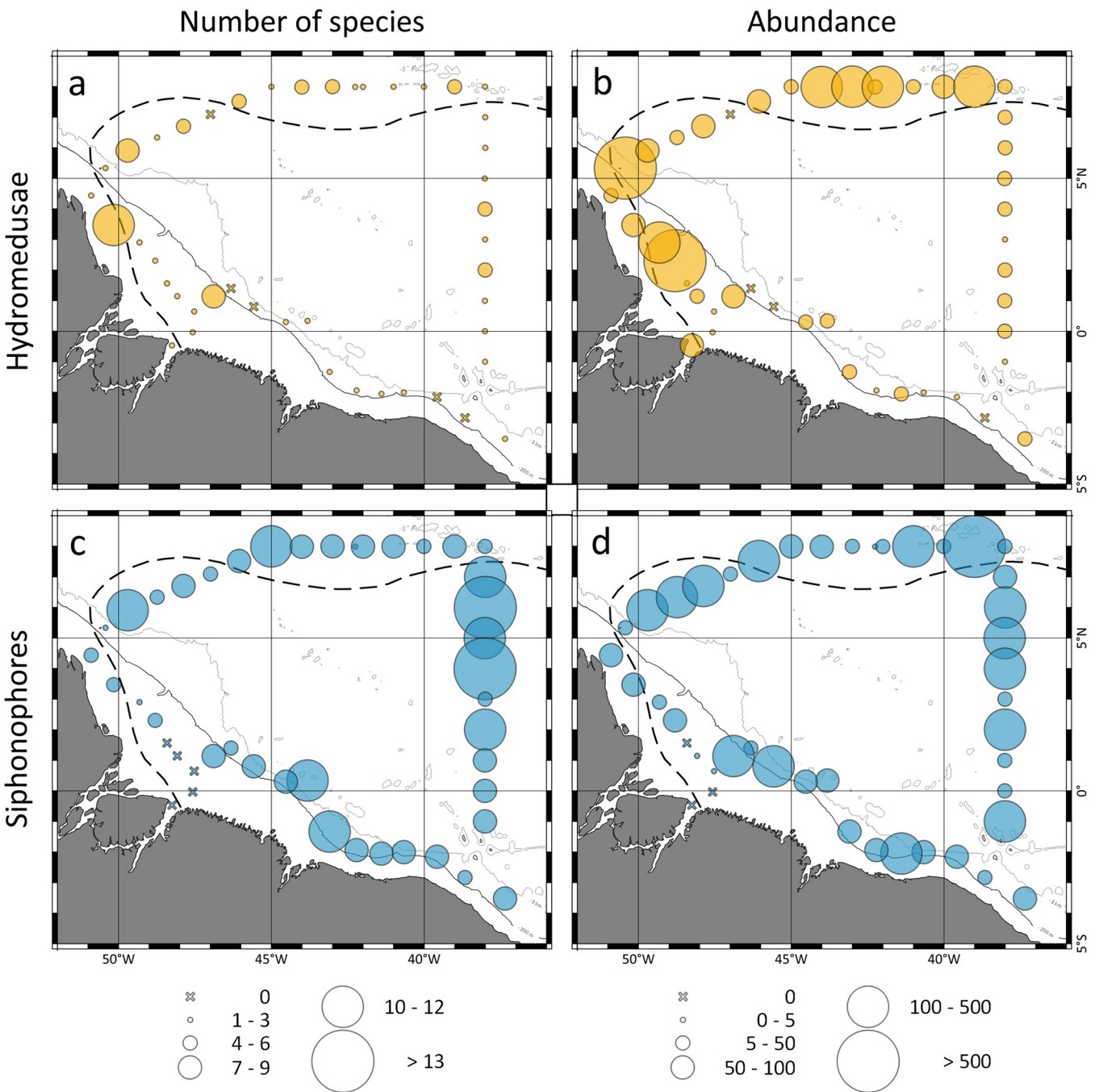

**Fig 2. Geographic distribution of number of species and total abundance (ind./or col. 100 m$^{-3}$) of hydromedusae (orange) and siphonophores (blue) found in the neustonic layer at October 2012.**

group B by oceanic stations. Exceptions were two stations located near the shelf break; station 10 is an oceanic station placed in group A and station 45 a neritic station placed in group B (Fig 4). The SIMPROF analysis indicated statistical significance for several subgroups in both main groups (Fig 4A). For practical purpose, the five small subgroups in the right branch of groups B were considered a single subgroup (B3).

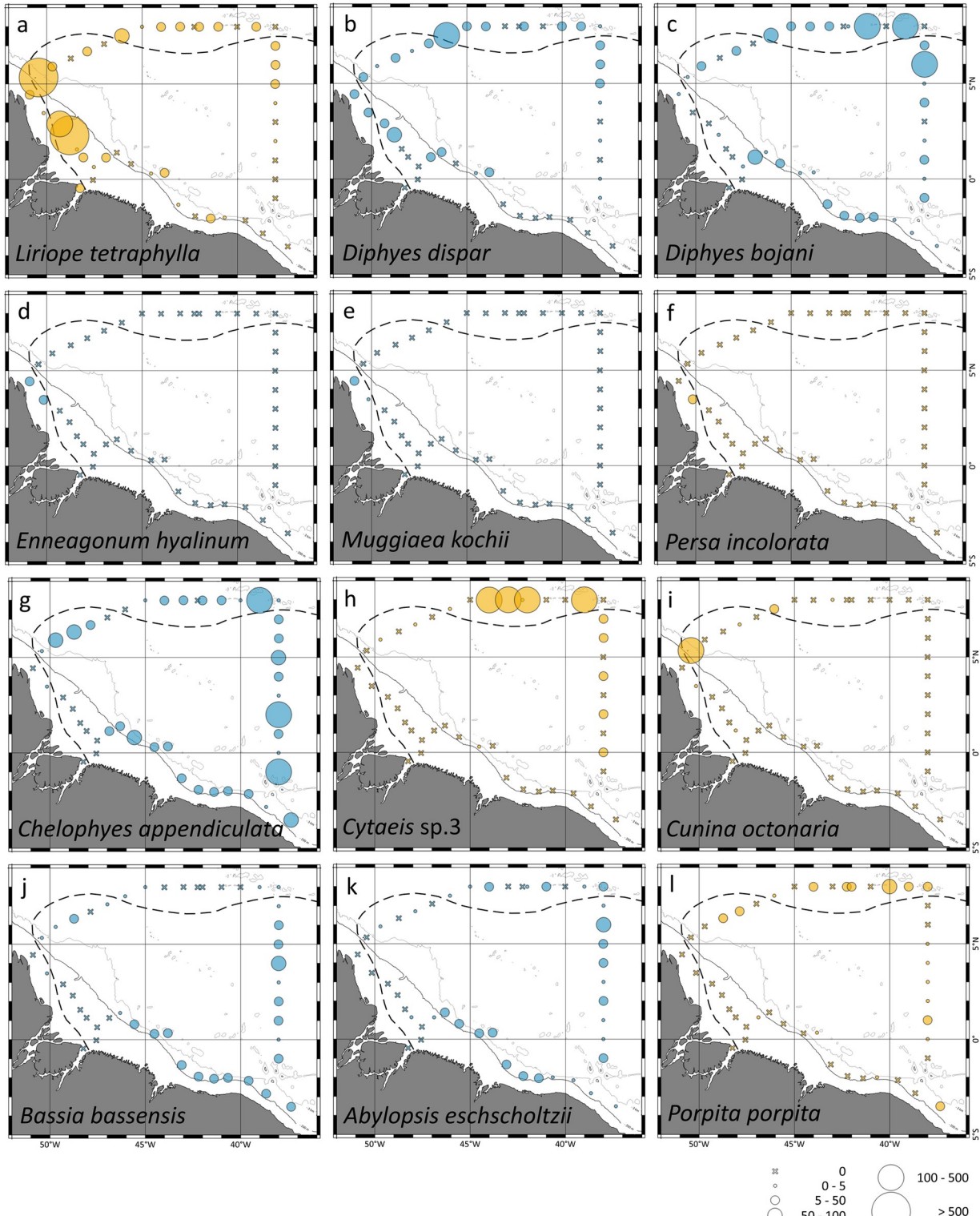

**Fig 3. Geographic distribution and abundance of representative hydromedusae (orange) and siphonophores (blue) species found in the neustonic layer at October 2012.**

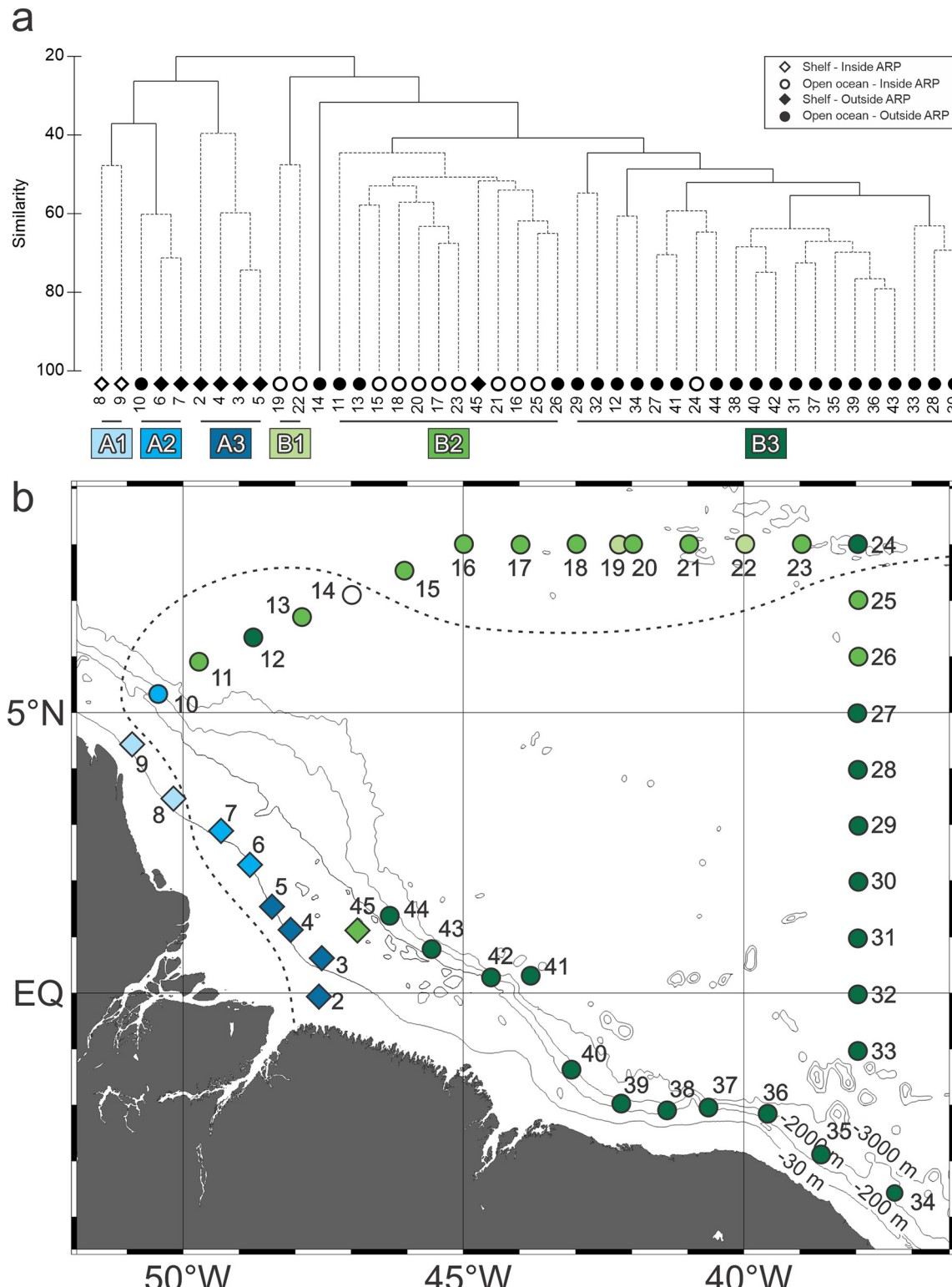

**Fig 4.** (a) Cluster analysis dendogram indicating two main groups and subgroups of stations with similar neustonic cnidarian communities in the Western Equatorial Atlantic Ocean. (b) Geographic distribution of cluster groups.

**Table 2. Results of SIMPER analysis, showing the relative contribution of neustonic cnidarian species in the formation of the groups defined in the cluster analysis.**

| Species | A1 | A2 | A3 | B1 | B2 | B3 |
|---|---|---|---|---|---|---|
| *Abylopsis eschscholtzi* | | | | | 7.59 | 16.36 |
| *Abylopsis tetragona* | | | | | | 9.6 |
| *Agalma okeni* | | | | | 3.48 | 3.77 |
| *Aglaura hemistoma* | | | | | 2.05 | |
| *Bassia bassensis* | | | | | | 19.81 |
| *Chelophyes appendiculata* | | | | | 13.39 | 21.7 |
| *Cytaeis* sp.3 | | | | | 8.39 | |
| *Diphyes bojani* | | | | | 20.01 | 9.11 |
| *Diphyes dispar* | 26.12 | 29.46 | | | 9.1 | |
| *Enneagonum hyalinum* | 33.44 | | | | | |
| *Eudoxoides mitra* | | | | | 1.78 | |
| *Liriope tetraphylla* | 10.47 | 64.95 | 100 | | 14.98 | |
| *Muggiaea kochii* | 10.47 | | | | | |
| *Nanomia bijuga* | 19.48 | | | | 3.63 | |
| *Porpita porpita* | | | | 100 | 2.66 | 3.95 |
| *Sulculeolaria chuni* | | | | | 4.35 | 3.93 |

Subgroup A1 was formed by the stations over the continental shelf under influence of the ARP (Fig 4). With 45.2% similarity, the group was represented by the siphonophores *Enneagonun hyalinum*, *D. dispar*, *Nanomia bijuga*, *Muggiaea kochii* and the hydromedusae *L. tetraphylla* (Table 2). Group A2 (61.4% similarity), was represented by stations 6, 7 and 10 were *L. tetrephylla* occurred in high abundance. *D. dispar* also was representative in the group (Fig 4, Table 2). Group A3 (25.1% similarity) included the remaining station over the continental shelf (Fig 4), where *L. tetraphylla* occurred almost alone (Table 2), and only occasional catches of other species occurred.

In the open ocean, subgroup B1 (41.3% similarity) was represented by two stations under influence of the ARP where *Porpita porpita* occurred in high abundance (Fig 4, Table 2). Subgroup B2 (50% similarity) included remaining stations under influence of the ARP (except for station 24), some stations near the plume borders and station 45, over the continental shelf (Fig 4). These stations presented higher abundances of *D. bojani*, *L. tetraphylla*, *D. dispar* and *Cytaeis* sp.3 than the remaining oceanic stations (included in B3). Other species representative in the group were *C. appendiculata*, *A. eschscholtzii* and *S. chuni* (Table 2). Subgroup B3 (51.4% similarity) was mainly represented by oceanic stations outside the ARP with higher abundances of *C. appendiculata*, *B. bassensis*, *A. eschscholtzii* and *Abylopsis tetragona* (Fig 4, Table 2).

## Responses to mesoscale processes and environmental gradient

The four canonical axes of the RDA explained 32.6% of species variance (Table 3). Monte Carlo test showed that the first (F-ratio = 9.2, P-value = 0.002) and all four canonical axes together (F-ratio = 3.183, P-value = 0.002) were significant. Bottom depth was negatively related to axis 1 and positively related to axis 2. Salinity and NBC were negatively related to axis 2. Temperature was positively related to both axes. Fluorescence was positively related to axis 1 and negatively related to axis 2. NECC was positively related to axis 2 (Fig 5; Table 3).

*Abylopsis eschscholtzii*, *Agalma okeni*, *C. appendiculata*, *Cytaeis* sp.3, *D. bojani*, *S. chuni* and *P. porpita* were positively related to bottom depth. *Cytaeis* sp.3 and *P. porpita* were also related to NECC. *A. tetragona*, *B. bassensis* and *Eudoxoides spiralis* were positively related to salinity (Fig 5).

**Table 3. Summary of the Redundancy Analysis performed between neustonic cnidarian species to environmental gradients and mesoscale processes in the Equatorial Atlantic Ocean.**

|  | Axis 1 | Axis 2 | Axis 3 | Axis 4 |
|---|---|---|---|---|
| Eigenvalues | 0.199 | 0.087 | 0.024 | 0.016 |
| Species-environment correlations | 0.851 | 0.668 | 0.663 | 0.386 |
| **Cumulative variance (%):** |  |  |  |  |
| Of species data | 19.9 | 28.7 | 31 | 32.7 |
| Of species-environment relation | 58.5 | 84.2 | 91.2 | 96 |
| **Correlations of explanatory variables:** |  |  |  |  |
| Bottom depth | -0.4482 | 0.7753 | 0.4079 | -0.1676 |
| Surface temperature | 0.5137 | 0.7917 | 0.0211 | -0.2514 |
| Surface salinity | -0.3457 | -0.5303 | 0.2381 | 0.4036 |
| Fluorescence | 0.4066 | -0.2563 | -0.2494 | -0.0206 |
| NBC | -0.1054 | -0.922 | -0.124 | -0.3258 |
| NECC | -0.0096 | 0.9265 | -0.3458 | -0.0861 |

*Diphyes dispar* and *N. bijuga* were negatively related to salinity and positively with temperature with little relation to bottom depth. *L tetraplyla* was slightly related to temperature and florescence and *A. hemistoma* was neither related to salinity or bottom depth (Fig 5).

## Discussion

Our results show that the neustonic cnidarian assemblages do not reflect well the strong responses to habitat heterogeneity observed in the communities inhabiting deeper waters in the Western Equatorial Atlantic. Consequently, they are not good proxy for the overall epipelagic cnidarian assemblage. Furthermore, they do not present specific responses to the surface habitat, which is highly structured by the ARP. Therefore, both initial hypotheses postulated here were denied using pelagic cnidarians as a model. In contrast, in general, the abundance of neustonic cnidarians was strikingly low across the entire study domain when compared to the values observed in epipelagic plankton samples (Fig 6) [30]. The dominant cnidarian species in each of the habitats from the Western Equatorial Atlantic were similar both in neuston and epipelagic communities. Therefore, the neustonic individuals seems to represent a remaining trace from the core of their populations present in deeper layers (Fig 7). These results are consistent with the range-edge effect hypothesis, which postulates that, proximate to the limits of their geographic ranges, most populations tend occur in lower abundance [51–53]. Such effect has been mostly discussed in terrestrial and benthic marine systems, where habitat boundaries are easily recognisable [54–56]. However, to our knowledge, it remains unreported in pelagic systems, where habitat boundaries are less apparent. As a major ecotone in the pelagic seascape, the neuston represents the upper boundary of epipelagic species population vertical ranges. Therefore, range-edge effect is likely to affect their abundance, in a similar manner to that we observed in the neustonic cnidarian community. Consequently, ecological assumptions of epipelagic communities based exclusively on neuston samples would be compromised.

This issue is reflected in the spatial structure of the neustonic cnidarian community observed in the cluster analysis, which was quite untidy in relation to the spatial structure of physical processes. This contrasts with the well-delimited picture of the area shaped by the epipelagic communities [30]. Although the frequency of ecological studies involving pelagic cnidarians has increased expressively in recent decades, the discussion of the most accurate way to sample them is still in its infancy. Progress has been made in understanding the optimal mesh opening size for sampling in estuarine, neritic and oceanic systems [57, 58]. However,

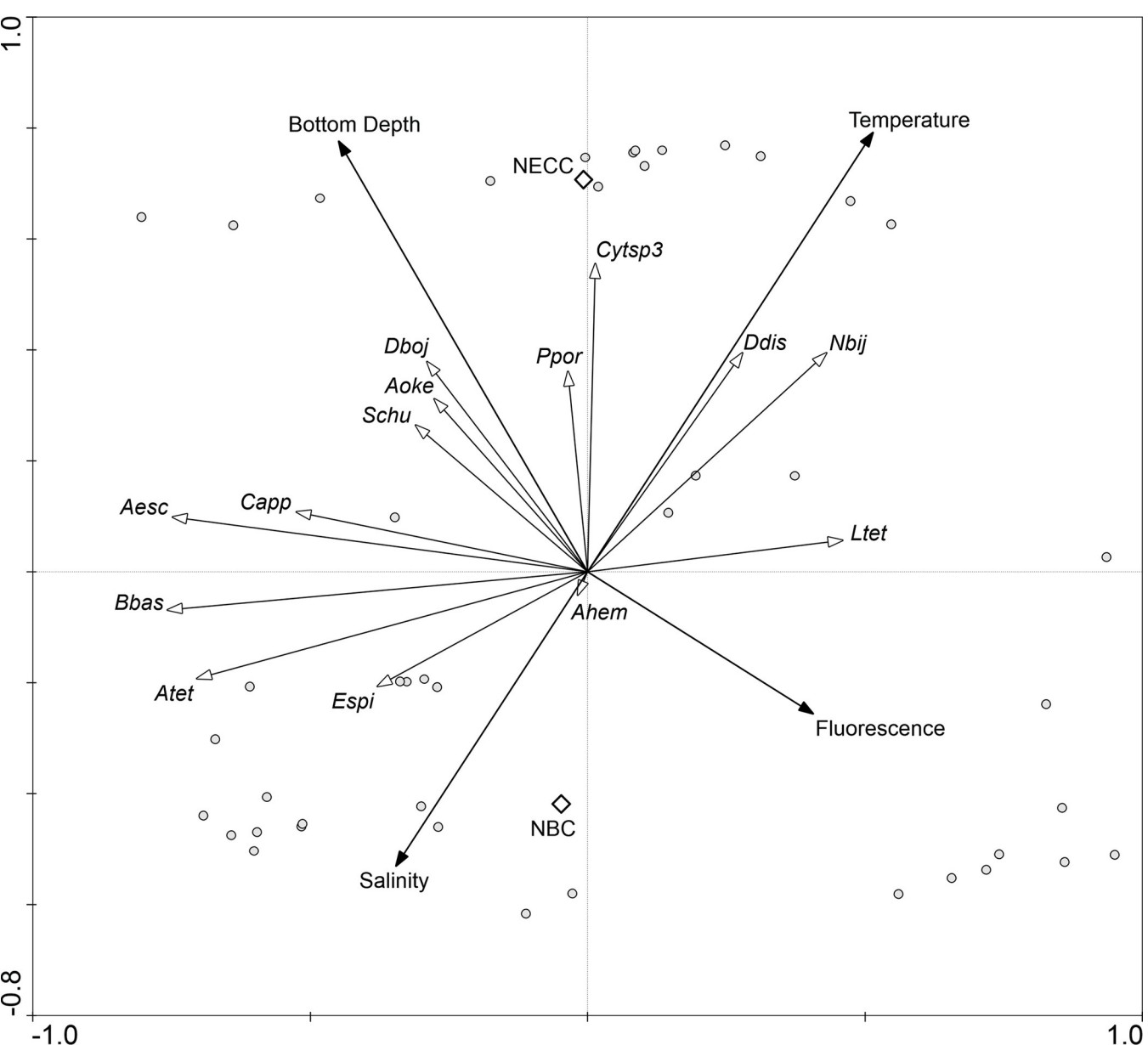

**Fig 5. Redundancy Analysis relating representative neustonic cnidarian species to environmental gradients and mesoscale processes in the Western Equatorial Atlantic Ocean.** NBC = North Brazilian Current; NECC = North Equatorial Counter Current. *Ltet = Liriope tetraphylla, Dboj = Diphyes bojani, Ahem = Aglaura hemistoma, Bbas = Bassia bassensis, Atet = Abylopsis tetragona, Nbij = Nanomia bijuga, Capp = Chellophyes appendiculata, Aesc = Abylopsis eschscholtzii, Schu = Sulculeolaria chuni, Ddis = Diphyes dispar, Aoke = Agalma okeni, Espi = Eudoxoides spiralis, Cytsp3 = Cytaeis* sp.3, *Ppor = Porpita porpita.*

the most appropriate layer of the water column to perform the tow in each type of habitat remains uncertain. Consequently, many ecological surveys have been based on subsurface (first tens of meters) trawls, particularly when sampling waters over continental shelves, but also in the open ocean [e.g. 59–62]. It can be reasonably assumed that the range-edge effect we observed in the neuston will also affect subsurface waters to a certain extent. Therefore, the outcomes from such sampling should be interpreted with caution, and a proper evaluation of spatial patterns in the structure of pelagic cnidarian communities should be rather grounded on stratified sampling.

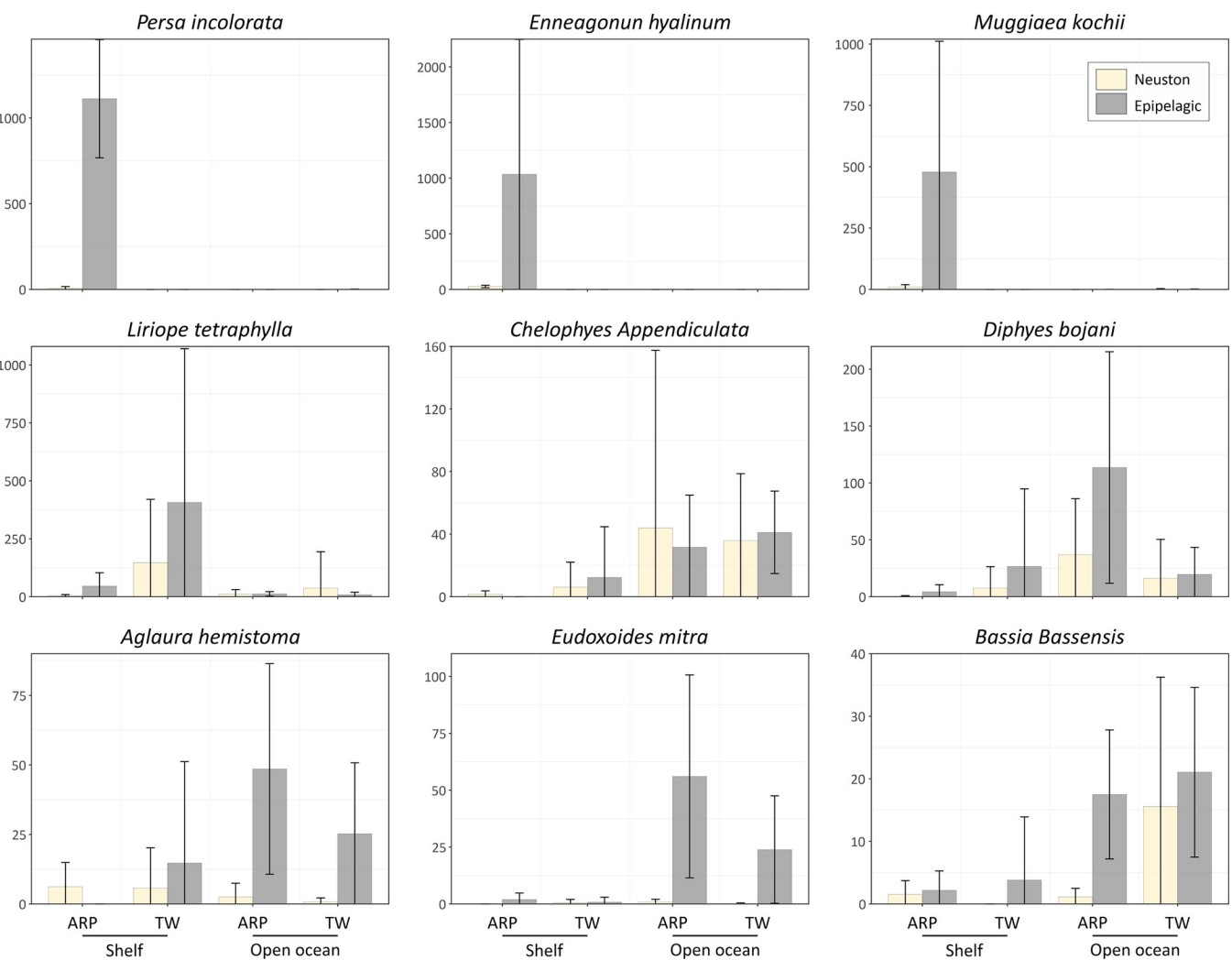

**Fig 6. Comparison of average abundance of representative cnidarian species in neustonic (yellow) and epipelagic (grey) layers, under influence of the Amazon River Plume (ARP) and Tropical Water (TW) over the continental shelf and in the open ocean in the Western Equatorial Atlantic Ocean.**

In the Western Equatorial Atlantic, the range-edge effect in the neustonic layer was more pronounced in the area under the influence of the ARP over the continental shelf. In this system, the dominant species, *E. hyalinun*, *M. kocchii*, *P. incolorata*, *D. dispar* and *N. bijuga*, occurred in quite larger abundances in the epipelagic strata, and were almost absent in the neuston (Fig 6). This finding is in contradiction with our hypothesis that the thin highly productive surface layer within the boundaries of the ARP would result in high diversity and abundance of neustonic cnidarians. Our results showed that, although the community was slightly more diverse in the habitats within the ARP over the continental shelf, the abundance of cnidarians was similar to that observed outside the plume, contrasting with the huge abundances observed in deeper layers in this system (Fig 6) [30].

In addition to the range edge effect, other processes may interplay. During the survey, below the thin (8 m depth) ARP surface layer, lied an oxygen minimum layer in consequence of the high rates of organic matter that sink and fuel microbial respiration [30, 63]. Such habitat structure is particularly beneficial for cnidarians, which usually are not restricted by oxygen concentration due to their low metabolic demand. However, the surface layer of the system is

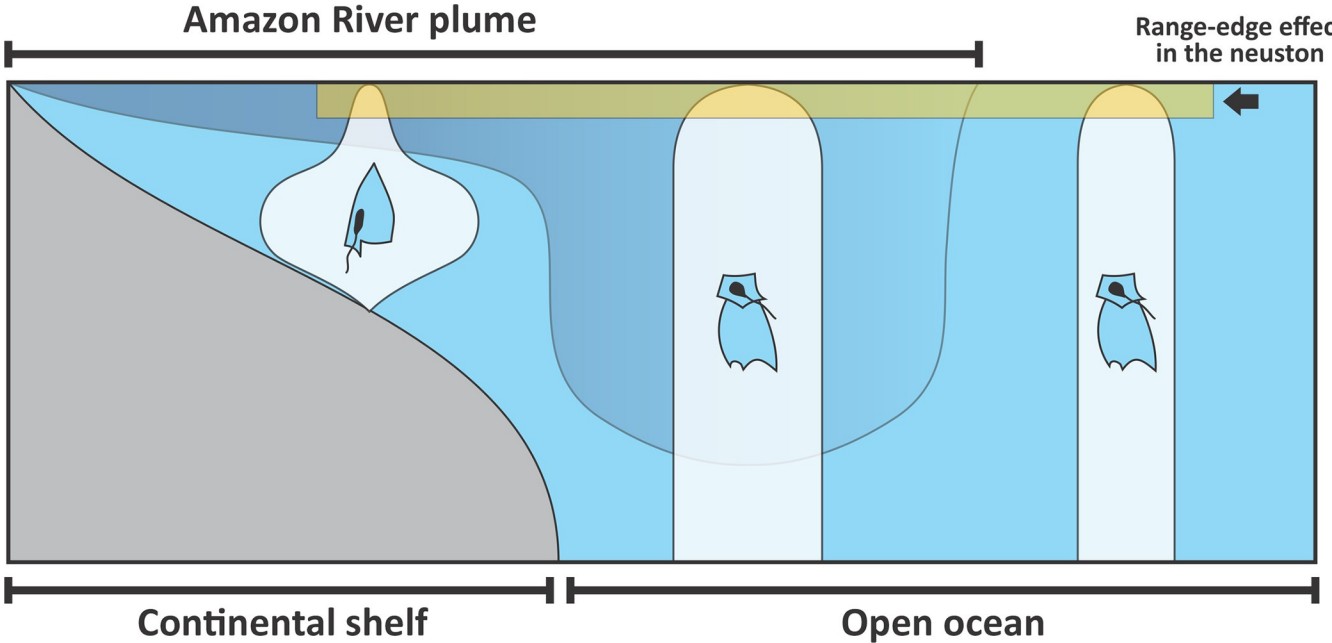

**Fig 7. Hypothesized scenarios of the distribution of pelagic cnidarian assemblages' abundance (not on scale) in the Western Equatorial Atlantic.**

well oxygenated, thus fish and other assemblages with potential to predate and compete with cnidarians are likely to be present in the neustonic community. The core of the cnidarian populations may thus be restricted to the oxygen minimum zone, fading towards the ARP until reaching the surface, where the range-edge effect was observed (Fig 6). Beyond species interactions, the low salinity of the ARP could restrict the populations of species present in the deeper, high-salinity layer from the neuston. However, the dominant species in deeper waters, are frequently found in shallow coastal and estuarine waters [24, 64–68]. This hypothesis is therefore less plausible. Additionally, the selectivity of the mesh used to sample the neustonic community, which was larger than the used in the epipelagic (500 vs. 300 and 120 μm) must be considered. However, while it indeed may have affected the sampling of small organisms such as *P. incolarata*, the remaining dominant species in this system were quite large, particularly *E. hyalinun* and *D. dispar* and likely well sampled with both gears.

Over the continental shelf outside the influence of the ARP and its oxygen minimum layer, where the entire water column was well mixed [30], the cnidarian assemblage in the nestonic and epipelagic [30] communities were more similar, being dominated by the hydromedusae *L. tetraphylla*, the only species which occurred in high abundance in both strata (Fig 6). However, in the neuston, the high abundances were restricted to stations located close to the boundaries of the ARP (Fig 3), likely being supported by the increased food supply in the vicinity. In contrast, in the epipelagic layer, the distribution was more evenly spread over the shelf.

Siphonophores, which are holoplanktonic, dominated the neustonic layer in the open ocean waters in number of species and abundance. This is to be expected, given that these organisms are not dependent on rigid substrates for polyp settlement, allowing them to survive and disperse offshore [69]. Beyond the siphonophores, the hydrozoans *L. tetraphylla*, *A. hemistoma*, *P. porpita* and *Cytaeis* sp.3 were representative in the open ocean as well. The two former species lack a polypoid stage in their life cycle and *P. porpita* is a pelagic hydroid stage, thus, these organisms are free to disperse offshore. However, anthoathecate hydrozoans such as *Cytaeis* sp.3 generally are meroplanktonic and its hydroids require rigid substrates [42].

Some factors may help explaining its distribution and high abundance in open waters: (i) The species was highly related to NECC, flowing eastward, thus this strong current may be carrying the medusae from shallow waters to the west, outside the range of our study area, such as the Caribbean Sea, where *Cytaeis* spp. are common [70, 71]; (ii) Most individuals collected presented medusae budding directly in their manubrium, what is common in the genus [40], this kind of asexual reproduction unbinds the species from the benthic stage and its demand for substrates; and (iii) the North Brazilian ridge, is present along the area [72], thus, several seamounts are present in the area could supply the dependence on substrates for hydroids of the species.

In the open ocean, similarly to the observed in the epipelagic communities [30], the ARP, with surface salinity as low as 32.5, did not cause significant changes in the species composition of neustonic cnidarian community and main divergences with the area outside the ARP occurred in species abundance. *D. bojani*, *D. dispar*, *P. porpita* and *Cytaeis* sp.3 were more abundant in the habitats under influence of the ARP. *D. bojani* and *D. dispar* were also among the dominant siphonophores in the epipelagic community of this habitat [30]. These two species have high niche plasticity, being present in neritic and oceanic habitats in a wide salinity range all over the world [24, 25, 62, 73, 74] and succeeded in the low salinity and highly productive environment of the ARP. *Eudoxoides mitra* and *Sminthea eurygaster* were dominant in the epipelagic community, but were almost absent in the neuston. These species are considerably large and therefore mesh selectivity is unlikely to have significantly affected the current estimated densities. Thus, we can infer that they preferentially inhabit the deeper layers where an oxygen minimum zone was also observed. Other siphonophore species present in offshore waters, such as *C. appendiculata*, *B. bassensis* and *A. escscholtzii* [75–78] did not present remarkable changes in abundance inside and outside the ARP, being widespread all over the oceanic waters of the study area indicating these species tolerate low salinity at least to some extent. Indeed, these species are common and eventually abundant at coastal waters [24, 68, 79] and occasionally found inside estuaries [80] along the Brazilian coast.

Although in the open ocean the general patterns in the neustonic and epipelagic cnidarian communities were similar, the differences observed are noteworthy. First, the already commented differences in species abundance between strata, which were quite lower in the neuston for most dominant species, are likely due the range-edge effect (Fig 7). Additionally, unlike the epipelagic community, in the neuston differences in species abundance inside and outside the plume were less prominent, more variable and did not follow exactly the 35 isohaline that characterize the ARP (Fig 4) [81]. This arrangement reflected the high variability in species abundance observed in the layer. Beyond range-edge effect other factors may be behind this variability. First, the neuston is sensitive to many oceanographic and atmospheric processes occurring in smaller scales than the ones evaluate in this study. Turbulence, submesocale eddies, waves, wind and variations in solar radiation caused by cloud cover and dial cycle may significantly contribute to plankton patchiness in sea surface [2, 82]. In addition, the volume of seawater filtered in neuston hauls is quite lower than in large bongo nets used to sample epipelagic, reducing chances of capturing patches of organisms, increasing variability, and reducing the probabilities of sampling rarer species.

Despite the failure to represent the patterns in the distribution of deeper assemblages' abundance, the neustonic layer of the continental shelf and adjacent oceanic waters in the area presented a diverse cnidarian community comprising 55 taxa. Hitherto, no other study was performed specifically considering the diversity of neustonic cnidarians, although 55 taxa may seem low when compared to the 91 taxa observed in epipelagic samples from the same area [30, 43], it is still remarkable since the neuston is a very thin portion of the water column, representing less than 0.2% of the depth sampled in the epipelagic hauls. In addition, many less

abundant species from the neuston were absent in deeper layers. Some of them are characteristic from the neuston and were already expected, such as the floating species *P. porpita*, *Physalia physalis* and *Velella velella* [83, 84]. Others were rare, such as *Stauridiosarsia producta* and *Pegantha laevis*, and its presence in the neuston may represent occasional catches. Differently, *Cytaeis* sp.3 occurred in high abundance exclusively in the neuston. This species does not have any apparent morphological adaptation for floating, such as pneumatophores or oil sacs, and its presence in the layer is perhaps due active behavior.

In this study, we investigated the intricate dynamics of the distribution and abundance of pelagic cnidarians in the neustonic layer in relation to the physical and ecological processes driven by the ARP in the Western Equatorial Atlantic. We conclude that the three-dimensional habitat structure induced by the spread of the Amazon River waters over the Western Equatorial Atlantic Ocean affects both the horizontal and vertical distribution of pelagic cnidarian assemblages. Over the continental shelf, the core of cnidarian species diversity and organisms' abundance seems to lie in the low oxygen zone under the ARP, since these organisms have low metabolic demand and are not restricted by oxygen. Meanwhile in the neuston, where oxygen is abundant and other interacting species are not restricted, abundance of cnidarian species was considerably lower than the observed in deeper layers. Beyond the influence of the plume, we conclude that species abundance is in general lower than in deeper water due the range-edge effect, which postulate that many populations close to the boundaries of their geographic ranges tend occur in lower abundance. Therefore, due the marked vertical structure observed here, proper evaluations of spatial patterns in the structure of pelagic cnidarian communities should preferentially be grounded on stratified sampling. Beyond the range-edge effect, the oceanographic and climatic instabilities occurring in the interface between ocean and atmosphere, are likely to affect their abundance.

## Supporting information

**S1 Data.**
(XLSX)

## Acknowledgments

We are grateful to the Brazilian National Institute of Science and Technology for Tropical Marine Environments, the Brazilian Research Network on Global Climate Change and European Integrated CARBOCHANGE for their support and to all the scientific team on board at the Camadas Finas III survey. We thank the CNPq (Brazilian National Council for Scientific and Technological Development). This work is a contribution to the LMI TAPIOCA (www.tapioca.ird.fr).

## Author Contributions

**Conceptualization:** Everton Giachini Tosetto, Sigrid Neumann-Leitão, Moacyr Araujo, Arnaud Bertrand, Miodeli Nogueira Júnior.

**Data curation:** Everton Giachini Tosetto, Sigrid Neumann-Leitão, Moacyr Araujo.

**Formal analysis:** Arnaud Bertrand, Miodeli Nogueira Júnior.

**Investigation:** Everton Giachini Tosetto, Sigrid Neumann-Leitão, Arnaud Bertrand, Miodeli Nogueira Júnior.

**Methodology:** Everton Giachini Tosetto, Moacyr Araujo, Arnaud Bertrand, Miodeli Nogueira Júnior.

**Project administration:** Sigrid Neumann-Leitão.

**Supervision:** Miodeli Nogueira Júnior.

**Visualization:** Arnaud Bertrand, Miodeli Nogueira Júnior.

**Writing – original draft:** Everton Giachini Tosetto.

**Writing – review & editing:** Sigrid Neumann-Leitão, Moacyr Araujo, Arnaud Bertrand, Miodeli Nogueira Júnior.

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
