## [Decision Letter · Decision Letter 0]

16 Oct 2024

PONE-D-24-29709Pelagic cnidarian assemblages show range-edge effect at the boundary of ocean surface, as illustrated in the case of the Amazon River PlumePLOS ONE

Dear Dr. Tosetto,

Thank you for submitting your manuscript to PLOS ONE. After careful consideration, we feel that it has merit but does not fully meet PLOS ONE’s publication criteria as it currently stands. Therefore, we invite you to submit a revised version of the manuscript that addresses the points raised during the review process.

The MS is valuable, well written and structured, reporting essential information regarding the abundance and distribution patterns of cnidarian assemblages in the neuston layer. There are only some minor concerns that should be fixed to make the paper fully published.

Please submit your revised manuscript by Nov 30 2024 11:59PM. If you will need more time than this to complete your revisions, please reply to this message or contact the journal office at plosone@plos.org. Please include the following items when submitting your revised manuscript:A rebuttal letter that responds to each point raised by the academic editor and reviewer(s). You should upload this letter as a separate file labeled 'Response to Reviewers'.A marked-up copy of your manuscript that highlights changes made to the original version. You should upload this as a separate file labeled 'Revised Manuscript with Track Changes'.An unmarked version of your revised paper without tracked changes. You should upload this as a separate file labeled 'Manuscript'.If applicable, we recommend that you deposit your laboratory protocols in protocols.io to enhance the reproducibility of your results. Protocols.io assigns your protocol its own identifier (DOI) so that it can be cited independently in the future. For instructions see: https://journals.plos.org/plosone/s/submission-guidelines#loc-laboratory-protocols. Additionally, PLOS ONE offers an option for publishing peer-reviewed Lab Protocol articles, which describe protocols hosted on protocols.io. Read more information on sharing protocols at https://plos.org/protocols?utm_medium=editorial-email&utm_source=authorletters&utm_campaign=protocols.

We look forward to receiving your revised manuscript.

Kind regards,

Claudio D'Iglio, Ph.D.

Academic Editor

PLOS ONE

Journal Requirements:

4. Your abstract cannot contain citations. Please only include citations in the body text of the manuscript, and ensure that they remain in ascending numerical order on first mention.

5. We note that [Figure 1] in your submission contain [map/satellite] images which may be copyrighted. All PLOS content is published under the Creative Commons Attribution License (CC BY 4.0), which means that the manuscript, images, and Supporting Information files will be freely available online, and any third party is permitted to access, download, copy, distribute, and use these materials in any way, even commercially, with proper attribution. For these reasons, we cannot publish previously copyrighted maps or satellite images created using proprietary data, such as Google software (Google Maps, Street View, and Earth). For more information, see our copyright guidelines: http://journals.plos.org/plosone/s/licenses-and-copyright.

Additional Editor Comments:

The MS of Tosetto et al. focused on the range-edge effect of cnidarian assemblage related to the ocean surface boundary. The paper covers enough aspects to interpret their findings, and their initial hypothesis rejected with the elaborated results. It is my opinion that the paper has a good structure, with a good amount of data coming from different sampling techniques. It requires only some minor adjustments to be suitoble for the publication.

Below just few specific comments.

Specific comments

Title:

The title is clear and well frames the performed study.

Abstract:

Abstract should summarize the performed study, presenting the importance of the study. In this case authors provide an exhaustive summary of the study performed and results obtained but it is missed the aim of the study, a possible ecological application of these findings.

Keywords:

Keywords are well sorted.

Introduction:

It is suggested to add a short insight related the zoology and morphology of the cnidarian groups, could be an addition information to better understand the ecologic role of these organisms. Ecology and data analysis aspects are fully explained with a good number of cited papers, well done.

Material and methods:

It is suggested to add some picture related to instruments used in order to better explain the sampling activity; studied area, species and relative abundance, software and all statistical tests are well explained.

Results:

Lines enumeration stop at 189 (pg. 8). In the “Responses to mesoscale processes and environmental gradient” section there are two statements related to bottom depth results, which is correct? Despite of these structure mistake, results are clearly presented and well organized considering the remarkable amount of data. Well done.

Discussion:

Results are good summarized and discussed in this section with a proper interpretation of study’s findings, also a coherent and updated bibliography is reported.

Reviewers' comments:

Reviewer's Responses to Questions

**Comments to the Author**

1. Is the manuscript technically sound, and do the data support the conclusions?

Reviewer #1: Yes

2. Has the statistical analysis been performed appropriately and rigorously? 

Reviewer #1: Yes

3. Have the authors made all data underlying the findings in their manuscript fully available?

Reviewer #1: Yes

4. Is the manuscript presented in an intelligible fashion and written in standard English?

Reviewer #1: Yes

5. Review Comments to the Author

Reviewer #1: The study conducted by Giachini Tosetto and colleagues focuses on the abundance and distribution patterns of cnidarian assemblages in the neuston layer, contributing valuable insights to marine ecology. The authors formulated clear hypotheses, providing well-supported results that challenge initial assumptions. The structure of manuscript follows a logical progression that is clearly explained from the introduction of the research questions to the presentation of results and their implications. From a scientific point of view, the focus on the interface between oceanographic and atmospheric processes is of high relevance, and the attention paid to the range-edge effects near the Amazon River Plume adds significant depth to our understanding of the spatial distribution of these communities. The fluent use of English and technical terms characterizes the text, as well as the clear and coincise language that makes the findings accessible to readers without oversimplifying the complex processes described. The discussion is well grounded in the results, especially regarding the suggested location of the community core in deeper hypoxic waters, which is an interesting conclusion. I would just like to suggest replacing Nausithoe aurea Silveira & Morandini, 1997 with the accepted name Nausithoe maculata Jarms, 1990 (according to the World Register of Marine Species - WoRMS).

6. PLOS authors have the option to publish the peer review history of their article (what does this mean?). If published, this will include your full peer review and any attached files.

Reviewer #1: No

---

## [Author Response · Author response to Decision Letter 0]

17 Oct 2024

Dear Prof. Dr. Claudio D'Iglio

Please find the corrected MS of our study “Pelagic cnidarian assemblages show range-edge effect at the boundary of ocean surface, as illustrated in the case of the Amazon River Plume”.

We sincerely appreciate the useful insights provided by you and the referee to improve our study. As detailed in the rebuttal letter below (responses in bold) we took into account all corrections and suggestions to prepare the revised version of our MS.

In relation to copyright of the maps stated in the decision email, the authors produced all maps specifically for this work with QGIS (stated in the methods) which is an open source software. Therefore, there is no need of additional permissions.

We hope that you will find that the revised manuscript adequately addressed all comments and that it now is suitable for publication in Plos ONE.

We thank you very much, in advance, for the attention you will grant to our re-submission.

Editor Comments:

The MS of Tosetto et al. focused on the range-edge effect of cnidarian assemblage related to the ocean surface boundary. The paper covers enough aspects to interpret their findings, and their initial hypothesis rejected with the elaborated results. It is my opinion that the paper has a good structure, with a good amount of data coming from different sampling techniques. It requires only some minor adjustments to be suitoble for the publication.

Below just few specific comments.

Title: The title is clear and well frames the performed study.

Abstract: Abstract should summarize the performed study, presenting the importance of the study. In this case authors provide an exhaustive summary of the study performed and results obtained but it is missed the aim of the study, a possible ecological application of these findings.

R. We explicitly stated the aim of the study in the resubmitted version of the MS.

Keywords: Keywords are well sorted.

Introduction: It is suggested to add a short insight related the zoology and morphology of the cnidarian groups, could be an addition information to better understand the ecologic role of these organisms. Ecology and data analysis aspects are fully explained with a good number of cited papers, well done.

R. Thanks for the suggestion, but the ecological relevance of cnidarians was already included in the introduction (lines 69-77)

Material and methods: It is suggested to add some picture related to instruments used in order to better explain the sampling activity; studied area, species and relative abundance, software and all statistical tests are well explained.

R. Thanks for the suggestion, we included a link to the instrument manual, which has also a picture, in the resubmitted version of the MS

Results: Lines enumeration stop at 189 (pg. 8). In the “Responses to mesoscale processes and environmental gradient” section there are two statements related to bottom depth results, which is correct? Despite of these structure mistake, results are clearly presented and well organized considering the remarkable amount of data. Well done.

R. Thanks for noticing, we corrected both.

Discussion: Results are good summarized and discussed in this section with a proper interpretation of study’s findings, also a coherent and updated bibliography is reported.

Reviewer Comments:

Reviewer #1: The study conducted by Giachini Tosetto and colleagues focuses on the abundance and distribution patterns of cnidarian assemblages in the neuston layer, contributing valuable insights to marine ecology. The authors formulated clear hypotheses, providing well-supported results that challenge initial assumptions. The structure of manuscript follows a logical progression that is clearly explained from the introduction of the research questions to the presentation of results and their implications. From a scientific point of view, the focus on the interface between oceanographic and atmospheric processes is of high relevance, and the attention paid to the range-edge effects near the Amazon River Plume adds significant depth to our understanding of the spatial distribution of these communities. The fluent use of English and technical terms characterizes the text, as well as the clear and coincise language that makes the findings accessible to readers without oversimplifying the complex processes described. The discussion is well grounded in the results, especially regarding the suggested location of the community core in deeper hypoxic waters, which is an interesting conclusion. I would just like to suggest replacing Nausithoe aurea Silveira & Morandini, 1997 with the accepted name Nausithoe maculata Jarms, 1990 (according to the World Register of Marine Species - WoRMS).

R. Thanks for the positive review. We corrected Nausithoe maculata name.

---

## [Editor Report · Decision Letter 1]

22 Oct 2024

Pelagic cnidarian assemblages show range-edge effect at the boundary of ocean surface, as illustrated in the case of the Amazon River Plume

PONE-D-24-29709R1

Dear Dr. Tosetto,

We’re pleased to inform you that your manuscript has been judged scientifically suitable for publication and will be formally accepted for publication once it meets all outstanding technical requirements.

Kind regards,

Claudio D'Iglio, Ph.D.

Academic Editor

PLOS ONE

---

## [Editor Report · Acceptance letter]

29 Oct 2024

PONE-D-24-29709R1 

PLOS ONE

Dear Dr. Tosetto, 

I'm pleased to inform you that your manuscript has been deemed suitable for publication in PLOS ONE. Congratulations! Your manuscript is now being handed over to our production team.

Kind regards, 

on behalf of

Dr. Claudio D'Iglio 

Academic Editor

PLOS ONE